# Enhanced Tomato Yellow Leaf Curl Thailand Virus Suppression Through Multi-Disease and Insect-Resistant Tomato Lines Combining Virus and Vector Resistance

**DOI:** 10.3390/insects16070721

**Published:** 2025-07-15

**Authors:** Shruthi Shimoga Prabhakar, Yun-Che Hsu, Joyce Yen, Hsiu-Yi Chou, Mei-Ying Lin, Mallapuram Shanthi Priya, Stephen Othim, Srinivasan Ramasamy, Assaf Eybishitz

**Affiliations:** 1Department of Genetics and Plant Breeding, Sri Venkateswara Agricultural College, Tirupati 517502, Andhra Pradesh, India; 2World Vegetable Center, Shanhua, Tainan 74199, Taiwanmei-ying.lin@worldveg.org (M.-Y.L.); srini.ramasamy@worldveg.org (S.R.)

**Keywords:** multi-disease and insect-resistant tomato, virus and vector resistance, TYLCTHV, whitefly, acylsugar, integrated pest management, *Solanum lycopersicum*

## Abstract

Tomatoes (*Solanum lycopersicum*) are highly vulnerable to the whitefly-transmitted tomato yellow leaf curl disease (TYLCD). This study evaluates multi-disease and insect-resistant tomato lines incorporating *Ty-1*/*Ty-3* genes (for virus resistance) and *WF2-10* and *WF3-09* genes (for whitefly resistance). Multi-disease and insect-resistant lines exhibit significantly higher acylsugar levels, which contribute to whitefly deterrence. These lines displayed reduced tomato yellow leaf curl Thailand virus (TYLCTHV) accumulation and milder disease symptoms over time. It was found that lines combining virus and vector resistance performed better than those with only Ty-resistance, whitefly resistance, or the susceptible control.

## 1. Introduction

Tomato, a globally important vegetable crop, is highly valued for its nutritional richness and culinary versatility. Tomato cultivation spans an area of approximately 5 million hectares worldwide, yielding a production of nearly 186.82 million tonnes with an average productivity of 36.97 tonnes per hectare [1].

Biotic stress in plants includes pests and diseases, which account for significant losses threatening crop yield and productivity [2]. The whitefly, *Bemisia tabaci* Gennadius (Hemiptera: Aleyrodidae), causes damage to the phloem and physiological disorders such as irregular fruit ripening, which reduces fruit quality and increases the number of unmarketable fruits [3]. Whiteflies excrete honeydew, which supports the growth of sooty mold fungus. However, its role as a plant virus vector causes the most serious damage [4]. Virus species belonging to five different genera (*Begomovirus*, *Crinivirus*, *Closterovirus*, *Ipomovirus*, and *Carlavirus*) are transmitted by whiteflies [5]. Among the viruses that can infect tomatoes, tomato yellow leaf curl virus (TYLCV) ranks third after tobacco mosaic virus and tomato spotted wilt virus on the list of the most important plant viruses worldwide [6]. TYLCV and at least 12 TYLCV-like *Begomovirus* species form a complex of viruses responsible for causing tomato yellow leaf curl disease (TYLCD) in various parts of the world. Tomato yellow leaf curl Thailand virus (TYLCTHV) is the predominant *Begomovirus* species responsible for TYLCD in Taiwan. TYLCTHV is mainly restricted to the tomato phloem tissues and is transmitted in a persistent circulative manner [7]. TYLCD causes leaf yellowing, curling, stunting, and, in severe cases, premature abscission of flowers and fruits, followed by cessation of plant growth [8]. Early tomato infection can cause up to 100% yield loss [9].

For profitable tomato production, TYLCD management is very important. Possible measures to control the spread and effects of the virus are breeding cultivars resistant to the virus and resistant to the vector. Resistance to TYLCD was found in several wild relatives of tomato, from which six virus resistance genes (*Ty-1* to *Ty-6*) have been identified. These genes have been mapped in wild tomato species including *S. chilense* (*Ty-1*, *Ty-3*, *Ty-4* and *Ty-6*), *S. habrochaites* syn. *L. hirsutum* (*Ty-2*), and *S. peruvianum* (*ty-5*). *Ty-1* and *Ty-2* genes express complete or nearly complete dominance, while *Ty-3* shows partial dominance [9]. The *ty-5* gene is recessively inherited and results from a loss-of-function mutation [8]. *Ty-1* and *Ty-3* are the primary resistance genes widely used in tomato breeding programs. The *Ty-1*/*Ty-3* gene combination conferred the highest level of resistance based on disease incidence and severity [9]. *Ty-2* is also used in breeding, either alone or in combination with other Ty-genes [8]. Ty genes have been successful in generating TYLCD-resistant commercial tomato cultivars, but generally have only reduced symptom severity, and the resistance achieved has never been 100% [10]. Plant co-infection by multiple *Begomovirus* strains offers opportunities to recombine and evolve new virulent, resistance-breaking forms of the virus. Another disadvantage of relying solely on Ty genes is that they constitute virus sources for susceptible genotypes [11].

While chemical vector control is ineffective, host plant resistance to insect vectors is suitable for managing circulative vector-borne viral diseases. Durable whitefly resistance, especially in the field, is more likely if tomato cultivars mount resistance based on a combination of antixenosis and antibiosis factors, thus forcing whiteflies to surmount a wide range of plant defenses [12]. Physically, plants deploy trichome and acylsugar-based strategies to restrain whiteflies from feeding. There are two groups of trichomes: glandular trichomes (Type I, IV, VI, and VII trichomes), which have “heads” containing various sticky and/or toxic exudates and secrete acylsugars, and non-glandular trichomes (Type II, III, and V trichomes) that do not secrete acylsugars. Acylsugars confer resistance to a wide range of important pests [12,13,14]. Natural resistance to multiple pests based on secretions by the different types of glandular trichomes present on stems and leaves of tomato and its wild relatives has been described [15]. Resistance is primarily through feeding deterrence, although oviposition deterrence also plays a role for certain pests. The capacity to repel and avoid whitefly landing, probing, and feeding is important to thwart whiteflies as vectors of plant viruses, especially *Begomoviruses*.

However, vector resistance alone would only lead to reduced virus spread, while the virus can still accumulate in plants without any virus resistance mechanism to slow down its replication. Thus, a holistic approach involving integrated pest management (IPM) is required for the management of TYLCD. IPM can include, along with other measures, host plant resistance to both the whitefly and the virus. Dual resistance in tomato cultivars would be valuable in repelling whiteflies and inhibiting virus replication in the plant, thus helping preserve the durability of virus resistance genes and possibly contributing to slowing *Begomovirus* evolution.

WorldVeg has developed BC_4_F_5_ breeding lines that are multi-disease and insect-resistant. The accession *VI007099* of *S. galapagense*, a close wild relative of the cultivated tomato, is resistant to the whitefly because it possesses type IV glandular trichomes that produce acylsucroses [12]. This resistance trait was introgressed into the multi-disease resistant elite line *CLN3682C,* and after four steps of recurrent backcrossing with selection for two whitefly resistance markers and acylsugar quantification, breeding lines *AVTO2428*, *AVTO2432*, *AVTO2437*, and *AVTO2436* were selected from *CLN4636BC4F5*. These lines contain *Ty-1*/*Ty-3* genes for virus resistance and *WF2-10* and *WF3-09* genes for whitefly resistance. The tomato lines carry resistance that confers protection against a broad spectrum of viruses, including TYLCTHV, which is predominant in Taiwan.

So far, limited research has examined the response of insect-resistant lines to the accumulation and spread of viruses causing TYLCD [16,17,18,19]. The reaction of dual-resistant lines to the virus accumulation has not been studied extensively. It remains unclear whether dual resistance against the insect vector and the virus limits *Begomovirus* accumulation because vector resistance can, in some cases, increase virus transmission [20,21]. However, vector-resistant cultivars have helped reduce the spread of other plant viruses [22]. The specific virus–vector interactions that determine *Begomovirus* transmission are complex, involving the virus and vector, the host plant, and the environment.

Therefore, this study was conducted to compare the TYLCTHV accumulation in plants combining virus and insect resistance with those resistant to either the insect or the virus alone and susceptible plants.

## 2. Materials and Methods

The experiment was performed under greenhouse conditions (26.7 ± 3.4 °C temperature; 69.1 ± 3.9% relative humidity) at WorldVeg in Tainan, Taiwan, from 23 February to 22 May 2024.

### 2.1. Tomato Plants, Virus Isolate, and Whitefly Population

Four multi-disease and insect-resistant lines, *AVTO2428*, *AVTO2432*, *AVTO2437*, and *AVTO2436* (*S. lycopersicum*); one virus-resistant line, *AVTO2445* (*S. lycopersicum*) (stable and advanced line of the recurrent female parent of the multi-disease and insect-resistant lines); one whitefly-resistant check, *AVTO2446* (*S. galapagense*); and one susceptible check, *AVTO9304* (*S. lycopersicum*), were used. The four multi-disease and insect-resistant lines are BC_4_F_5_ lines developed by transferring insect resistance from *S. galapagense* accession into an elite multiple disease-resistant line (including TYLCD resistance). The characteristics of the lines used in the study are summarized in Table 1.

Tomato plants infected with TYLCTHV were obtained from the virology department of WorldVeg. In this experiment, we used the LJ3-5 isolate of TYLCTHV, originally collected from Kaohsiung, Taiwan. The complete genomic sequences of this isolate, including both DNA-A and DNA-B components, are publicly available in GenBank under accession numbers EF577266 and EF577267. Healthy *B. tabaci* individuals (both Q and B biotypes) were obtained from the WorldVeg virology department and reared on cabbage plants (*Brassica oleracea* var. *capitata*; “Green Tide” variety) in insect cages in an insect-proof plastic house. Custom-built insect cages (100 cm × 60 cm × 70 cm) enclosed with 50-mesh insect-proof netting were used. Viruliferous whiteflies were obtained by releasing healthy adults on TYLCTHV-infected tomato plants and allowing them to feed for four days.

### 2.2. Whitefly Preference, Adult Mortality Bioassay, and TYLCTHV Control Inoculation

A choice bioassay was conducted to study the whitefly preference. The photographs depicting the choice assay experimental setup used to evaluate plant preference by adult whiteflies is present in Appendix A. The 28-day-old seedlings of three plants of each of the 7 genotypes were placed randomly in a circle within insect cages, and the setup was replicated thrice. A total of 70 *B. tabaci* adult mating pairs were released into the cage, resulting in 10 mating pairs per genotype. The viruliferous whiteflies were released in the center of the circle. The inoculation access period (IAP) was 8 days (192 h). After the IAP, the whitefly adults were counted on the whole plants. Adult mortality (%) was calculated by taking the percentage of the number of dead whiteflies found on the plant to the total number of whiteflies on the plant. The plants were then treated with Confidor insecticide.

### 2.3. TYLCTHV Accumulation

TYLCTHV was detected by semi-quantitative Polymerase Chain Reaction (PCR). The DNA extraction procedure was performed according to Fulton et al. with modifications [23]. A detailed step-by-step protocol for DNA extraction from tomato leaf samples is provided in Appendix A. The genomic DNA was quantified through Nanodrop (Thermo Scientific NanoDrop spectrometer, Waltham, MA, USA) and then diluted to 10 ng/μL. The diluted sample was subjected to PCR (Biorad CFX96 Real Time PCR System, Bio-Rad Laboratories, Inc., Hercules, CA, USA). Each plant was sampled and tested for the presence and amount of TYLCTHV in the plants (3-, 5- and 20-day post-inoculation) by comparing with the standard curve developed using serially diluted plasmid DNA at different cycles as shown in Figure 1.

To assess relative viral load, PCR was conducted for each sample at four different cycle numbers: 15, 20, 25, and 30. Amplified products were run through agarose gel electrophoresis, and band presence or absence was assessed. This was compared against a standard curve generated using serial dilutions of plasmid DNA (1, 10^−1^, 10^−2^, 10^−3^, and 10^−4^ ng/μL). Scored electrophoresis data is in Appendix A and raw gel images are available in Appendix A. As shown in Figure 1, plasmid DNA at 1 ng/μL yielded visible bands at all four cycle numbers, while lower concentrations showed amplification at progressively higher cycles. Samples showing bands at all four cycles were interpreted to contain at least 1 ng/μL of viral DNA. PCR amplification was carried out in a 10 μL reaction volume containing 6.0 μL of double-distilled water, 1.0 μL of 10× PCR buffer, 0.6 μL of dNTPs (Premix dNTP, 2.5mM; Protech Technology Enterprise Co., Ltd., Taipei City, Taiwan), 0.1 μL each of forward (GGACATGCAGGTGAGGAGTCC) and reverse primers (TTATACGGATGGCCGCTTT), 0.2 μL of Taq DNA polymerase (Super-Therm Gold DNA Polymerase; JMR Holdings Inc., West Midlands, UK), and 2.0 μL of DNA template. The thermal cycling conditions consisted of an initial denaturation at 95 °C for 10 min, followed by 15, 20, 25, or 30 cycles of denaturation at 95 °C for 30 s, annealing at 60 °C for 45 s, and extension at 72 °C for 45 s. A final extension step was performed at 72 °C for 5 min.

### 2.4. Disease Scoring

The plants were phenotypically evaluated and scored based on the severity of the disease symptoms using a 1–6 scale as shown in Figure 2.

### 2.5. Acylsugar Assay

A standard peroxidase/glucose oxidase (PGO)-based acylsugar assay was performed following the methodology described by Savory [24]. The protocol for acylsugar quantification in plant samples is provided in Appendix A.

### 2.6. Statistical Analysis

Data were subjected to statistical analysis using SPSS for Windows, version 25.0 (SPSS Inc., Chicago, IL, USA). The data were checked for normality and variance homogeneity using Shapiro–Wilk and Levene’s tests, respectively. A Sqrt(x) transformation was applied to normalize the adult whitefly data, whereas an arcsine(x) transformation was applied to adult mortality (%). Analysis of variance (ANOVA) was performed, and the means were compared through Tukey’s Honestly Significant Difference (HSD) test (*p* < 0.05). The Kruskal–Wallis test was performed for acylsugar content, virus accumulation, and disease severity index, followed by Dunn’s test (*p* < 0.05) for pairwise comparison wherever statistically significant differences were found.

## 3. Results

A series of controlled greenhouse experiments were conducted to assess the response of multi-disease and insect-resistant tomato lines to the accumulation of TYLCTHV. Summary of the experimental results are provided in Appendix A. The results are summarized below.

### 3.1. Acylsugar Content in Leaves

The total acylsugar concentration in different genotypes is presented in Figure 3. The total acylsugar concentration was lower in the genotypes *AVTO2445* and *AVTO9304*. It was highest in *AVTO2428*, followed by *AVTO2432*, *AVTO2437*, *AVTO2436*, and *AVTO2446*. A Kruskal–Wallis test indicated a significant difference in acylsugar content across the seven genotypes (df = 6, N = 21; test statistic χ^2^ = 14.892, *p* = 0.021). The acylsugar concentrations of *AVTO2445* and *AVTO9304* were not significantly different from each other but were significantly lower than the multi-disease and insect-resistant lines, viz., *AVTO2428*, *AVTO2432*, and *AVTO2437*.

### 3.2. Whitefly Preference Assay

The genotype *AVTO2446* attracted the fewest whitefly adults in the choice assay as shown in Figure 4. A significant difference was found between genotypes [F(6,14) = 3.745, *p* = 0.020], with *AVTO2446* (M = 0.74, SD = 0.65) differing significantly from *AVTO9304* (M = 2.55, SD = 0.48). However, *AVTO2428*, *AVTO2432*, *AVTO2437*, *AVTO2436*, and *AVTO2445* did not differ significantly from one another, or from *AVTO2446* or *AVTO9304*. Adult mortality (%) was also higher in the multi-disease and insect-resistant lines (Figure 5).

### 3.3. TYLCTHV Accumulation

Figure 6 summarizes the viral load on various days post-inoculation. The multi-disease and insect-resistant plants consistently displayed lower viral loads than *AVTO2445*, *AVTO2446*, and *AVTO9304*. At 20 days post-inoculation (dpi), the genotype *AVTO2432* had significantly lower virus amounts than *AVTO2445*, *AVTO2446*, and *AVTO9304*.

### 3.4. Disease Scoring

Figure 7 summarizes the disease scoring results. The multi-disease and insect-resistant plants consistently showed lower disease scores and none to very mild symptoms. At 35 dpi, the effectiveness of multi-disease and insect-resistant plants is demonstrated by an average severity rating of 1, indicating that all plants were healthy. In contrast, single-resistant plants and the susceptible check had average severity ratings of 2.50 and 4.44, respectively. At 35 dpi, all four lines differed significantly in disease severity from *AVTO2445* and *AVTO9304*.

## 4. Discussion

This study demonstrates that combining resistance to *B. tabaci* based on acylsugar secretion with virus resistance genes (*Ty-1*/*Ty-3*) in multi-disease and insect-resistant tomato lines significantly reduced the accumulation of the TYLCTHV compared to plants carrying either insect or virus resistance alone and, as expected, to the susceptible check.

The choice bioassay is expected to reveal lines that confer both antixenosis and antibiosis resistance mechanisms [25]. The initial host plant selection by adults is primarily influenced by preference factors. The preference assays confirmed that the lines with high acylsugar secretion like *AVTO2428*, *AVTO2432*, *AVTO2437*, *AVTO2436*, and *AVTO2446* were less preferred, whereas lines *AVTO2445* and *AVTO9304* were more preferred. Previous studies have suggested that glandular trichomes on leaves and acylsugar secretion were associated with reduced attractiveness and resistance to *B. tabaci* [12,15,16,17,26,27,28]. Our results are aligned with these findings, with lines showing higher acylsugar concentration being less preferred by whiteflies. Whitefly feeding, besides vectoring viruses, causes a range of other disorders. The multi-disease and insect-resistant lines might help to reduce the irregular ripening disorder caused by *B. tabaci* on tomatoes by lowering whitefly infestations [17]. Host resistance to whiteflies presumably also minimizes the spread of other viruses transmitted by whiteflies, such as tomato chlorosis virus [29], thereby lowering the risk of the emergence of recombinant viruses due to mixed infection. Due to the combination of different resistance mechanisms, multi-disease and insect-resistant lines might also exert lower selection pressure on the virus to evolve [30].

Resistance to adult *B. tabaci* in the tomato lines was associated with high acylsugar concentration in these lines, which reduced the plant’s attractiveness to the insect and, therefore, reduced settling and oviposition. Rodríguez-López et al. (2011) [17] found that the insect-resistant plants showing deterrence due to trichomes and acylsugar concentrations alter the feeding behavior of the insect after it lands on a plant and affect virus acquisition by decreasing the ability to start probing. The tested whitefly resistance did not completely block virus acquisition and transmission in vector-resistant plants but significantly reduced the initial viral concentration. Hence, combining vector resistance with virus resistance to reduce disease development is necessary.

The graph of virus accumulation over time indicated that when the whiteflies were given a free choice between the genotypes, more whiteflies were attracted to the susceptible check *AVTO9304*. This leads to high initial virus inoculation. Coupled with the lack of any virus resistance mechanism to reduce viral multiplication, higher virus accumulation was seen in the susceptible check. In the virus-resistant genotype *AVTO2445*, the viral titer was found to be as high as in the susceptible genotypes at the initial stages, due to whitefly-mediated inoculation. However, despite the high initial viral load, further virus multiplication appears to be suppressed, likely due to the presence of *Ty-1*/*Ty-3* resistance genes. This suggests that the genes do not prevent initial infection but play a significant role in limiting viral replication and reducing symptom severity. In the vector-resistant genotype *AVTO2446*, the initial virus concentration is low due to whitefly resistance, but due to the absence of any mechanism to reduce viral multiplication, even a small initial viral load multiplies at a high rate, resulting in high virus accumulation over time and leading to disease development. In contrast, the multi-disease and insect-resistant lines consistently showed lower virus concentrations. At 20 dpi, the multi-disease and insect-resistant line *AVTO2432* performed significantly better than the single-resistant plants and the susceptible check. This could be attributed to the combined resistance targeting the virus and its vector, creating a more comprehensive barrier against TYLCTHV infection and accumulation. These cultivars deter whitefly infestation and exhibit significant efficacy in impeding viral proliferation within the plant tissues.

Lower viral loads corresponded with reduced symptom severity, demonstrating the effectiveness of multi-disease and insect resistance in alleviating disease symptoms. The mild symptoms observed in the dual-resistant plants can be attributed to their enhanced ability to restrict virus replication and movement within the plant tissues and reduced virus acquisition potential.

In conclusion, this study demonstrates that the multi-disease and insect-resistant tomato lines significantly reduce the accumulation of TYLCTHV and disease symptoms. The high acylsugar concentration reduces the preference of whiteflies for settling, resulting in reduced initial viral load. Together with virus resistance, this leads to lower virus accumulation over time and less severe symptoms, underscoring the effectiveness of these lines in managing TYLCTHV. These lines offer a robust and sustainable solution to prevent TYLCD damage in tomato breeding programs and can reduce the reliance on chemical controls to combat disorders caused by whiteflies. The study highlights the potential of multi-disease and insect-resistant tomato lines as a key component of IPM strategies to manage whitefly-transmitted TYLCTHV effectively. These lines are a breakthrough in resistance breeding for TYLCD management. The promising results from this study pave the way for further research and development of multi-disease and insect-resistant lines to combat TYLCD and other whitefly-transmitted viral diseases.

Further research should focus on conducting field trials under diverse environmental conditions and with different virus complexes causing TYLCD to validate the effectiveness of these lines in real-world agricultural settings, exploring the long-term durability of resistance and the potential for resistance breakdown. Studies should be conducted to assess the impact on fruit production, both in controlled conditions comparing controls and inoculated samples, and in field conditions. Further, the effect of pyramiding insect resistance with other Ty resistance genes and resistance genes for other viruses of tomato into these multi-disease and insect-resistant lines needs to be investigated to optimize the implementation of these resistant lines as an IPM component to maximize their benefits in agricultural practices. Further research should also focus on improving the horticultural traits such as fruit size and yield according to the market demands for better adoption by farmers.

## Figures and Tables

**Figure 1 insects-16-00721-f001:**
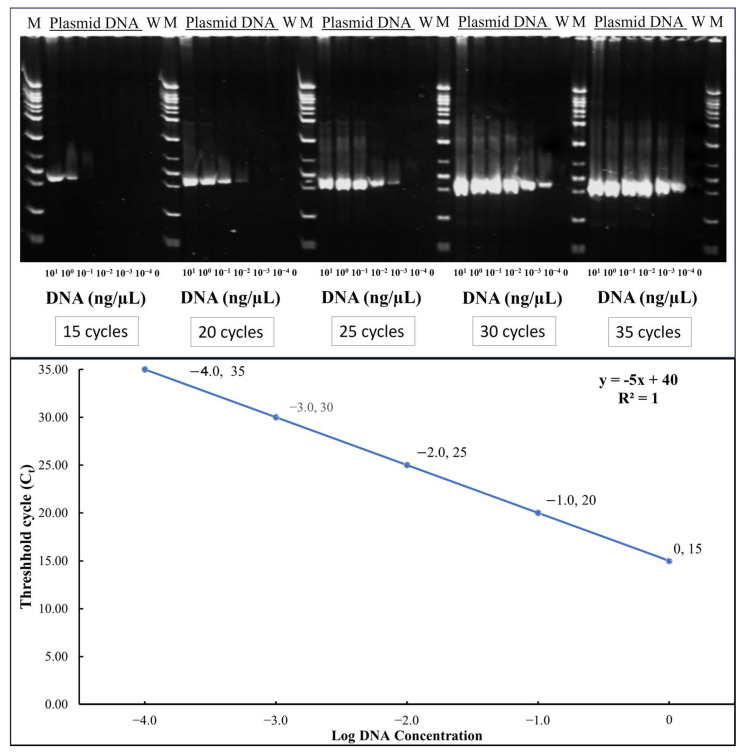
Gel electrophoresis of PCR products of serially diluted plasmid DNA at different cycles and standard curve: threshold cycles(C_t_) vs. log DNA concentration (y = −5x + 40). The values shown represent the log-transformed DNA concentration with conversions as follows: Log −4.0 = 0.0001 ng/μL; Log −3.0 = 0.001 ng/μL; Log −2.0 = 0.01 ng/μL; Log −1.0 = 0.1 ng/μL; Log 0.0 = 1 ng/μL.

**Figure 2 insects-16-00721-f002:**
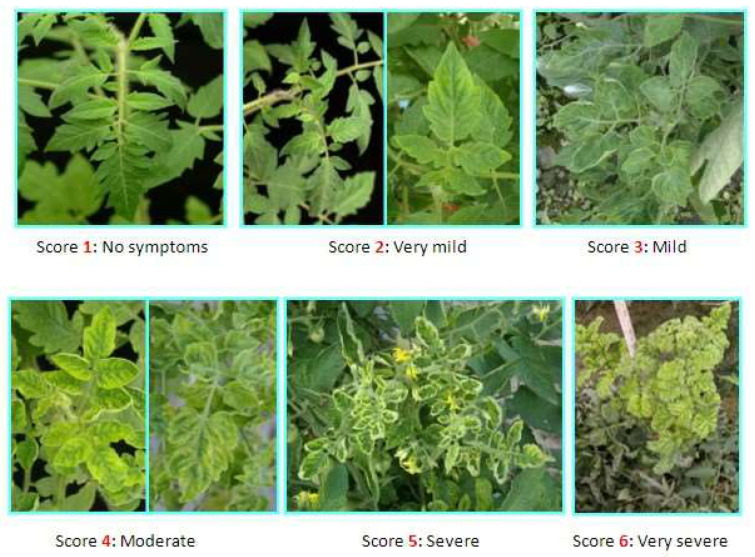
Visual representation of the 1–6 phenotyping scale for scoring disease severity in plants. 1 = Healthy; 2 = Very mild; 3 = Mild; 4 = Moderate; 5 = Severe; 6 = Very severe.

**Figure 3 insects-16-00721-f003:**
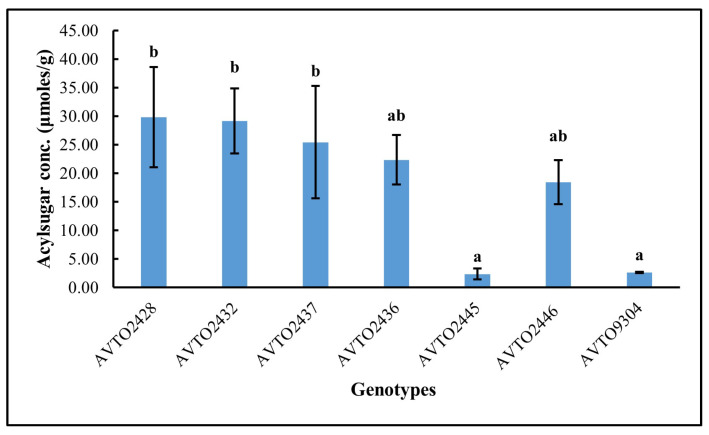
Total acylsugar concentration (μmoles/g) across genotypes. Different letters indicate statistical differences according to Dunn’s post hoc tests (*p* < 0.05). Error bars represent standard deviation.

**Figure 4 insects-16-00721-f004:**
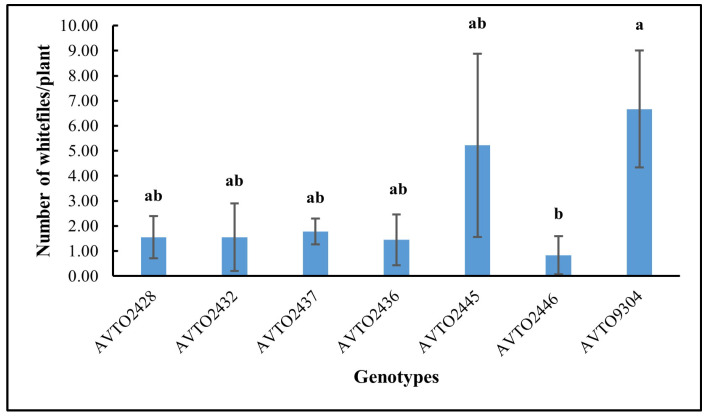
Average number of adult whiteflies per plant across genotypes. Different letters indicate statistical differences according to Tukey’s HSD test (*p* < 0.05). Error bars represent standard deviation.

**Figure 5 insects-16-00721-f005:**
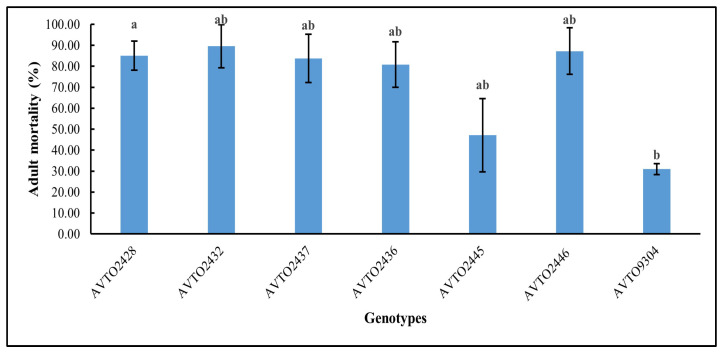
Percentage of adult whitefly mortality in different plant genotypes. Different letters indicate statistical differences according to Tukey’s HSD test (*p* < 0.05). Error bars represent standard deviation.

**Figure 6 insects-16-00721-f006:**
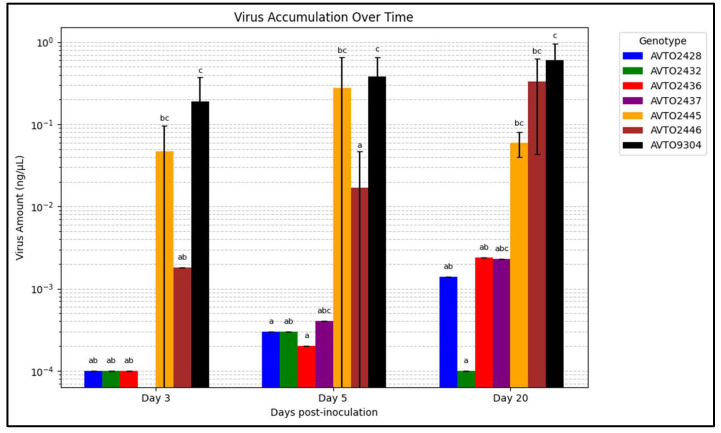
Graph illustrating the change in virus accumulation over time. Different letters indicate statistical differences according to Dunn’s tests (*p* < 0.05). Error bars represent standard deviation.

**Figure 7 insects-16-00721-f007:**
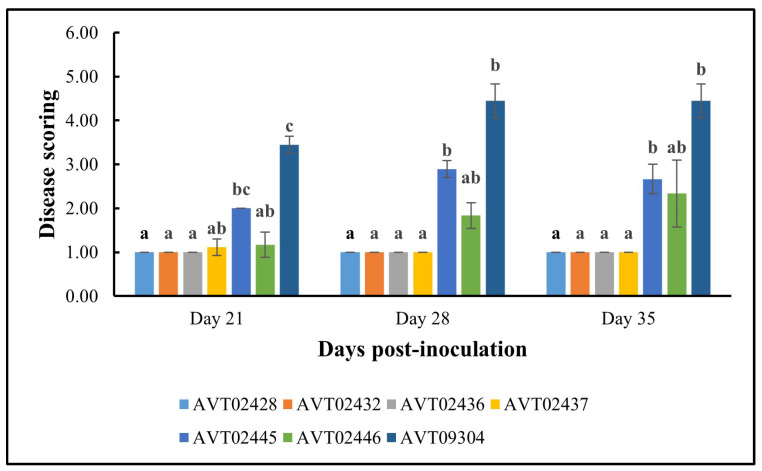
Results of disease severity scores for different plant genotypes using a 1–6 scale. 1 = Healthy; 6 = Very severe. Different letters indicate statistical differences according to Dunn’s tests (*p* < 0.05). Error bars represent standard deviation.

**Table 1 insects-16-00721-t001:** Characteristics of lines used in the study.

Genotype	Pedigree	Characteristic	Ty1/3	WF2-10	WF3-09
AVTO2428	CLN4636BC4F2-31I03-1(a)-BK(e)	Multi-disease and insect-resistance line	+	+	+
AVTO2432	CLN4636BC4F2-31I03-1(d)-BK(e)	Multi-disease and insect-resistance line	+	+	+
AVTO2436	CLN4636BC4F2-31I03-1(c)-1(d)	Multi-disease and insect-resistance line	+	+	+
AVTO2437	CLN4636BC4F2-31I03-1(c)-4(a)	Multi-disease and insect-resistance line	+	+	+
AVTO2445	CLN4396F1-2-18Z85-23-16-9-6	Virus-resistant line	+	−	−
AVTO2446	VI063177-10 (BL1921)	Whitefly-resistant line	−	+	+
AVTO9304	CL5915-93D4-1-0-3	Susceptible check	−	−	−

+ Presence of resistance allele; − Absence of resistance allele.

## Data Availability

The original contributions presented in this study are included in the article/Appendix A. Further inquiries can be directed to the corresponding author.

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
