# Peer review of "Enhanced Tomato Yellow Leaf Curl Thailand Virus Suppression Through Multi-Disease and Insect-Resistant Tomato Lines Combining Virus and Vector Resistance"

_insects, 2025, doi:10.3390/insects16070721_

Round 1
Reviewer 1 Report
Comments and Suggestions for Authors
Dear authors;
Your article titled “Enhanced TYLCV Suppression through Multi-Disease and Insect-Resistant Tomato Lines Combining Virus and Vector Resistance” is generally well structured, supported by detailed experimental data, and fills an important gap in the literature. I kindly request you to make the minor corrections in the file in the system.
My best regards

Author Response
Response to Reviewer 1 Comments
|
||
1. Summary |
|
|
Thank you very much for taking the time to review this manuscript. Please find the detailed responses below and the corresponding revisions/corrections in track changes in the re-submitted files.
|
||
2. Questions for General Evaluation |
Reviewer’s Evaluation |
|
Does the introduction provide sufficient background and include all relevant references? |
Yes |
|
Are all the cited references relevant to the research? |
Yes |
|
Is the research design appropriate? |
Yes |
|
Are the methods adequately described? |
Yes |
|
Are the results clearly presented? |
Yes |
|
Are the conclusions supported by the results? |
Yes |
|
3. Point-by-point response to Comments and Suggestions for Authors |
||
Comments 1: Please, write lower case T (L14). |
||
Response 1: Thank you for pointing this out. We agree with this comment. Therefore, the suggested correction was done. [“tomato yellow leaf curl”- L14] |
||
Comments 2: Please, write lower case “Tomato Yellow Leaf Curl Virus” (L22) |
||
Response 2: Thank you for bringing this to our attention. We agree and have updated the manuscript accordingly. [“tomato yellow leaf curl virus”- L22] Comments 3: Please, write italics - Bemisia tabaci (L49) Response 3: Thank you for highlighting this. We agree with this comment. Therefore, the suggested correction was done. [“Bemisia tabaci Gennadius”- L48] |
||
4. Response to Comments on the Quality of English Language |
||
Point 1: The English is fine and does not require any improvement. |
||
Response 1: Thank you for your positive feedback regarding the quality of the English language. |
Reviewer 2 Report
Comments and Suggestions for Authors
This paper presents promising results on combining vector and virus resistance to control tomato yellow leaf curl virus in tomatoes. The approach is interesting and appears to be effective based on the data provided.
The introduction is well written and clearly sets up the context and goals of the study. I’d also like to commend the authors for including detailed methods in the Supplementary Materials, which supports transparency and reproducibility. That said, the supplementary files for results are much disorganized and are not necessary. I was also unable to access Supplementary Figures 1 and 2.
I’ve made a few minor comments and suggestions in the attached file, mostly about the large number of figures and tables, some of which are excessive.
Once these points are addressed, I believe the manuscript will be ready for publication.

Author Response
Response to Reviewer 2 Comments
|
|
|||
1. Summary |
|
|
|
|
Thank you very much for taking the time to review our manuscript. Please find below the detailed point-by-point responses. The corresponding revisions have been made and are highlighted using track changes in the re-submitted files.
|
|
|||
2. Questions for General Evaluation |
Reviewer’s Evaluation |
|
||
Does the introduction provide sufficient background and include all relevant references? |
Yes |
|
||
Are all the cited references relevant to the research? |
Yes |
|
|
|
Is the research design appropriate? |
Can be improved |
|
|
|
Are the methods adequately described? |
Can be improved |
|
|
|
Are the results clearly presented? |
Can be improved |
|
|
|
Are the conclusions supported by the results? |
Can be improved |
Materials and methods, results and conclusions have been revised and refined. |
|
|
3. Point-by-point response to Comments and Suggestions for Authors |
|
|
||
Comments 1: This paper presents promising results on combining vector and virus resistance to control tomato yellow leaf curl virus in tomatoes. The approach is interesting and appears to be effective based on the data provided. The introduction is well written and clearly sets up the context and goals of the study. I’d also like to commend the authors for including detailed methods in the Supplementary Materials, which supports transparency and reproducibility. That said, the supplementary files for results are much disorganized and are not necessary. I was also unable to access Supplementary Figures 1 and 2. I’ve made a few minor comments and suggestions in the attached file, mostly about the large number of figures and tables, some of which are excessive. |
|
|||
Response 1: Thank you for your positive and encouraging feedback. We appreciate your thoughtful comments and suggestions. We have revised the supplementary materials by removing the results file and ensuring all supplementary figures are now properly included and accessible. We have removed few tables to avoid redundancy. Suggestions and comments from the annotated file have also been addressed, and corresponding changes are made in the revised version using track changes. |
|
|||
Comments 2: Replace TYLCV with tomato yellow leaf curl virus. (L2) |
|
|||
Response 2: Thank you for pointing this out. We agree with this comment. Therefore, the suggested correction was done. [“Tomato Yellow Leaf Curl Virus”- L2] Comments 3: Whitefly-transmitted tomato yellow leaf curl virus (TYLCV) (L13) Response 3: Thank you for highlighting this. We agree with this comment. Therefore, the suggested correction was done [“the whitefly-transmitted tomato yellow leaf curl virus (TYLCV)”- L13-L14]. |
|
|||
Comments 4: Lower case “Tomato Yellow Leaf Curl Virus” (L22). |
|
|||
Response 4: Thank you for pointing this out. We agree with this comment. Therefore, the suggested correction was done. [“tomato yellow leaf curl virus”- L22] |
|
|||
Comments 5: a virus transmitted by whiteflies (L23) |
|
|||
Response 5: Thank you for your observation. We agree with your comment and have made the recommended correction accordingly. [“a virus transmitted by whiteflies”- L23] Comments 6: Strikethrough IPM (L37) Response 6: Thank you for pointing this out. We agree with this comment. Therefore, the suggested correction was done. [“sustainable Integrated Pest Management strategies”- L37] |
|
|||
Comments 7: Add Solanum lycopersicum in keywords (L39). |
|
|||
Response 7: We are grateful for your suggestion. The suggested correction was done. [“Solanum lycopersicum”- L40] |
|
|||
Comments 8: Italics - Bemisia tabaci (L49) |
|
|||
Response 8: Thank you for pointing this out. The suggested correction was done. [“Bemisia tabaci”- L48] Comments 9: Include species to which TYLCV belongs (L57) Response 9: Thank you for your comment. In our study, we focused on Tomato yellow leaf curl Thailand virus (TYLCTHV), the most prevalent begomovirus species affecting tomatoes in Taiwan. TYLCTHV belongs to the genus Begomovirus, family Geminiviridae, and is part of the Tomato yellow leaf curl virus species complex. |
|
|||
Comments 10: Are all of these partial/quantitative resistances? Clarify (L66). |
|
|||
Response 10: Thank you for pointing this out. We have revised the text to clarify the nature of resistance conferred by each gene. [“These genes have been mapped in wild tomato species including S. chilense (Ty-1, Ty-3, Ty-4 and Ty-6), S.habrochaites syn. L. hirsutum (Ty-2), and S. peruvianum (ty-5). Ty-1 and Ty-2 genes express complete or nearly complete dominance, while Ty-3 shows partial dominance. The ty-5 gene is recessively inherited and resulted from a loss-of-function mutation. Ty-1 and Ty-3 are the primary resistance genes widely used in tomato breeding programs. The highest resistance as based on disease incidence and severity was provided by Ty-1/Ty-3 gene combination.”- L66-L72] |
|
|||
Comments 11: acylsugars, (L83) |
|
|||
Response 11: Thank you for pointing this out. We agree with this comment. Therefore, the suggested correction was done. [“acylsugars,”- L86] Comments 12: qPCR instead of PCR (L152) Response 12: Thank you for pointing this out. However, semi-quantitative endpoint PCR, instead of qPCR, was performed in this study. Therefore, the original text was retained. Comments 13: Strikethrough the and virus (L109) Response 13: Thank you for pointing this out. We agree with this comment. Therefore, the suggested correction was done. [“accumulation and spread of TYLCV”- L112] |
|
|||
Comments 14: Instead of was not include has not been (L110). |
|
|||
Response 14: Thank you for pointing this out. We agree with this comment. Therefore, the suggested correction was done. [“virus accumulation has not been studied extensively.”- L113] |
|
|||
Comments 15: Italics- S. galapagense (L127) |
|
|||
Response 15: Thank you for pointing this out. The suggested correction was done. [“S. galapagense”- L130] Comments 16: Where were the dead whiteflies found? on the plant? (L149) Response 16: Thank you for pointing this out. We have revised the text to provide clarity. [“Adult mortality (%) was calculated by taking the percentage of the number of dead whiteflies found on the plant to the total number of whiteflies on the plant.”- L151-L153] |
|
|||
Comments 17: I am very confused about the methodology for the viral quantification – was qPCR performed, or were bands visually compared in a gel? This section needs better writing. (L151). Response 17: Thank you for your comment. Quantitative PCR (qPCR) was not performed in this study. Instead, we used semi-quantitative endpoint PCR to estimate viral DNA accumulation. PCR amplification was carried out for each sample at four different cycle numbers: 15, 20, 25, and 30 cycles. The amplified products were separated by agarose gel electrophoresis, and the presence or absence of bands at different cycle numbers was visually assessed. This was compared with the standard dilution series of plasmid DNA with known concentrations to estimate relative viral DNA levels. As shown in Figure 1, plasmid DNA at 1 ng/μL yielded visible bands at all four cycle numbers, while lower concentrations showed amplification at progressively higher cycles. We have revised the 2.3. TYLCV virus accumulation section to better reflect this approach and ensure clarity. (L154-L182) |
|
|||
Comments 18: Fulton et al. with modifications (L153) |
|
|||
Response 18: Thank you for pointing this out. We agree with this comment. Therefore, the suggested correction was done. [“Fulton et al. with modifications [23].”- L156-L157] Comments 19: Nanodrop (L155) Response 19: Thank you for pointing this out. The suggested correction was done. [“Nanodrop”- L158] |
|
|||
Comments 20: qPCR (L157). |
|
|||
Response 20: Thank you for pointing this out. However, semi-quantitative endpoint PCR, instead of qPCR, was performed in this study. Therefore, the original text was retained. |
|
|||
Comments 21: Figure 1 is unnecessary (L162) |
|
|||
Response 21: Thank you for your suggestion. However, we believe that Figure 1 enhances the clarity of the methodology used for viral quantification. Therefore, we have retained the figure in the revised manuscript. Comments 22: It is unusual to see PCR reaction mixture and conditions in Tables. It would be more appropriate to have them described in the text. (L167) Response 22: Thank you for pointing this out. We agree with the comment and have revised the manuscript accordingly. The PCR reaction mixture and thermal cycling conditions have been removed from the table and are now described in the main text. [“PCR amplification was carried out in a 10 μL reaction volume containing 6.0 μL of double-distilled water, 1.0 μL of 10× PCR buffer, 0.6 μL of dNTPs (10 mM each), 0.1 μL each of forward and reverse primers, 0.2 μL of Taq DNA polymerase, and 2.0 μL of DNA template. The thermal cycling conditions consisted of an initial denaturation at 95°C for 10 minutes, followed by 15, 20, 25, or 30 cycles of denaturation at 95°C for 30 seconds, annealing at 60°C for 45 seconds, extension at 72°C for 45 seconds, and a final extension step at 72°C for 5 minutes.”- L176-L182]. |
|
|||
Comments 23: Peroxidase/glucose oxidase (PGO) (L181). |
|
|||
Response 23: Thank you for pointing this out. The suggested correction was done. [“Peroxidase/glucose oxidase (PGO)”- L192] |
|
|||
Comments 24: Define HSD (L190) |
|
|||
Response 24: Thank you for pointing this out. I agree with this comment. Therefore, the suggested correction was done. [“Tukey’s Honestly Significant Difference (HSD) test”- L201] Comments 25: Strikethrough the and various (L219) Response 25: Thank you for highlighting this. We agree with this comment. Therefore, the suggested correction was done. [“Average number of adult whiteflies per plant across genotypes.”- L231] |
|
|||
Comments 26: Strikethrough the and was observed (L222). |
|
|||
Response 26: We are grateful for your suggestion. We agree with this comment. Therefore, the suggested correction was done. [“Percentage of adult whitefly mortality in different plant genotypes.”- L235] |
|
|||
Comments 27: Figure 6 is enough to depict these results. Table 5 is not needed (L225) |
|
|||
Response 27: We are grateful for your suggestion. We agree with this comment and the suggested correction was done. Comments 28: Keep formatting consistent between Figures 6 and 7 grouping bar graphs by time point in both. This would also solve an issue – the letters above bars in Figure 6 are relative to the time points, not to comparisons within genotypes, which is confusing. (L235) Response 28: Thank you for pointing this out. We appreciate your insightful feedback. We agree and the suggested correction was done. [Fig 6 was changed- L245] |
|
|||
Comments 29: Lowercase t in Tomato (L265). |
|
|||
Response 29: Thank you for pointing this out. We agree with this comment. Therefore, the suggested correction was done. [“as tomato chlorosis”- L278] |
|
|||
Comments 30: Rephrase line - The high initial virus concentration multiplies faster (L282) |
|
|||
Response 30: Thank you for your suggestion. The suggested correction was done. [“A higher initial concentration of virus accelerates the rate of viral multiplication”- L295] Comments 31: But this is not what is being shown -- viral titer in AVTO2445 is as high as in the susceptible genotypes. Do Ty-1 and Ty-3 reduce titer, or only symptoms? (L283) Response 31: Thank you for pointing this out. In the virus-resistant genotype AVTO2445, the viral titer was found to be as high as in the susceptible genotypes at initial stages, due to whitefly-mediated inoculation. However, despite the high initial viral load, further virus multiplication appears to be suppressed, likely due to the presence of Ty-1/Ty-3 resistance genes. This suggests that the genes do not prevent initial infection but play a significant role in limiting viral replication and reducing symptom severity. [L226-301] |
|
|||
Comments 32: It would be important to assess the effect of fruit production in future studies, both in controlled conditions comparing controls and inoculated samples, and in field conditions. Please include that in the discussion. (L315). |
|
|||
Response 32: Thank you for pointing this out. We agree with this comment. Therefore, the suggested correction was done. [“Further research should focus on conducting field trials under diverse environmental conditions and with different TYLCV strains to validate the effectiveness of these lines in real-world agricultural settings, exploring the long-term durability of resistance and the potential for resistance breakdown. Studies should be conducted to assess the impact on fruit production, both in controlled conditions comparing controls and inoculated samples, and in field conditions.”- L332-L337] |
|
|||
4. Response to Comments on the Quality of English Language |
|
|||
Point 1: The English is fine and does not require any improvement. |
|
|||
Response 1: Thank you. We are glad the language meets the required standard.
|
|
Reviewer 3 Report
Comments and Suggestions for Authors
1. Introduction
The introduction lacks a clear and cohesive structure, making it difficult to follow the main theme of the study. Revising the introduction for better organization, clarity, and language accuracy would significantly improve the manuscript.
2. Materials and Methods
Could you clarify how temperature and relative humidity were maintained within such a narrow range in the greenhouse? (L120-121)
Please provide the scientific name of the tomato and the scientific name and cultivar of the cabbage used in the study. Which biotype of Bemisia tabaci did you use for the experiment, MEAM1, MED, or another? What was its source? (L124-139)
In the introduction, you mention that the tomato lines are resistant to TYLCV, but the virus used in the study was TYLCTHV. TYLCV and TYLCTHV are considered distinct species. Are tomato lines resistant to TYLCV also resistant to TYLCTHV? (L134-139)
Please provide the isolate name of TYLCTHV used in the study. Has the genome sequence been published or made accessible? (L134-139)
The subheading does not fully reflect the content of the section, as you also included an experiment on adult mortality. (L140)
Please provide the information of the insect cages and insecticide used in the study. (L150)
Real-time PCR is recommended for virus quantification. Tables 2 and 3 could be integrated into the main text of the section. Additionally, please provide details of the PCR kit used and the primer sequences. (152-169)
3. Results
Do Table 5 and Figure 6 originate from the same dataset? Including either one would be sufficient.
Figures 3, 4, 5, 6, and 7. What do the error bars represent?
Author Response
Response to Reviewer 3 Comments
|
|
||||
1. Summary |
|
|
|
||
Thank you very much for taking the time to review our manuscript. We appreciate your insightful comments and suggestions, which has helped us to improve the quality of the manuscript. Below, we have provided detailed, point-by-point responses to each comment.
|
|
||||
2. Questions for General Evaluation |
Reviewer’s Evaluation |
|
|||
Does the introduction provide sufficient background and include all relevant references? |
Must be improved |
|
|||
Are all the cited references relevant to the research? |
Must be improved |
|
|
||
Is the research design appropriate? |
Must be improved |
|
|
||
Are the methods adequately described? |
Can be improved |
|
|
||
Are the results clearly presented? |
Yes |
|
|
||
Are the conclusions supported by the results? |
Can be improved |
Introduction, materials and methods, and conclusions have been revised and refined. |
|
||
3. Point-by-point response to Comments and Suggestions for Authors |
|
|
|||
Comments 1: The introduction lacks a clear and cohesive structure, making it difficult to follow the main theme of the study. Revising the introduction for better organization, clarity, and language accuracy would significantly improve the manuscript. |
|
||||
Response 1: Thank you for your suggestion. The introduction has been revised. Language has also been refined to enhance readability. (Not much changes done as there is conflicting reviews about the introduction. To better enhance readability, can subheadings can be given to introduction? ) |
|
||||
Comments 2: Could you clarify how temperature and relative humidity were maintained within such a narrow range in the greenhouse? (L120-121) |
|
||||
Response 2: Thank you for pointing this out. Temperature and relative humidity were maintained using an automated system. Sensor placed in the greenhouse recorded the temperature and humidity every one hour interval. The average temperature and humidity were calculated over the experimental period and mentioned in the materials and method. Comments 3: Please provide the scientific name of the tomato and the scientific name and cultivar of the cabbage used in the study. Which biotype of Bemisia tabaci did you use for the experiment, MEAM1, MED, or another? What was its source? (L124-139) Response 3: The tomato lines were developed by WorldVeg and are all derived from Solanum lycopersicum—they are advanced elite breeding lines, not wild types. The cabbage varieties used in the study are commercial cultivars from Known-You Seed, named "Green Tide" variety. |
|
||||
Comments 4: In the introduction, you mention that the tomato lines are resistant to TYLCV, but the virus used in the study was TYLCTHV. TYLCV and TYLCTHV are considered distinct species. Are tomato lines resistant to TYLCV also resistant to TYLCTHV? (L134-139) |
|
||||
Response 4: The tomato lines presented in this article carry resistance markers that confer protection against a broad spectrum of virus strains, including Tomato yellow leaf curl Thailand virus (TYLCTHV), the predominant strain in Taiwan. |
|
||||
Comments 5: Please provide the isolate name of TYLCTHV used in the study. Has the genome sequence been published or made accessible? (L134-139) |
|
||||
Response 5: In this experiment, we used the LJ3-5 isolate of Tomato yellow leaf curl Thailand virus (TYLCTHV), originally collected from Kaohsiung, Taiwan. The complete genomic sequences of this isolate, including both DNA-A and DNA-B components, are publicly available in GenBank under accession numbers EF577266 and EF577267. Comments 6: The subheading does not fully reflect the content of the section, as you also included an experiment on adult mortality. (L140) Response 6: Thank you for pointing this out. We agree with this comment. Therefore, the suggested correction was done. [“Whitefly preference, adult mortality bioassay and TYLCV control inoculation”- L143] |
|
||||
Comments 7: Please provide the information of the insect cages and insecticide used in the study. (L150). |
|
||||
Response 7: The cage was custom-built with dimensions of 100 × 60 × 70 cm and enclosed with 50-mesh insect-proof netting and Confidor insecticide. |
|
||||
Comments 8: Real-time PCR is recommended for virus quantification. Tables 2 and 3 could be integrated into the main text of the section. Additionally, please provide details of the PCR kit used and the primer sequences. (152-169) |
|
||||
Response 8: Thank you for your valuable suggestion. We acknowledge that real-time PCR offers greater sensitivity and precision for virus quantification. However, in the present study, semi-quantitative PCR was employed. Despite this, the method allowed us to successfully compare viral accumulation levels across treatments. As suggested, Tables 2 and 3 have now been integrated into the main text. PCR kit details and primer sequence? F: GGACATGCAGGTGAGGAGTCC R: TTATACGGATGGCCGCTTT (Can i include the primer sequences?) Comments 9: Do Table 5 and Figure 6 originate from the same dataset? Including either one would be sufficient. Figures 3, 4, 5, 6, and 7. What do the error bars represent? Response 9: Thank you for your valuable suggestion. In the revised manuscript, we have retained only Figure 6. The error bars in Figures 3, 4, 5, 6, and 7 represent the standard deviation. |
|
||||
4. Response to Comments on the Quality of English Language |
|
||||
Point 1: The English could be improved to more clearly express the research. |
|
||||
Response 1: Thank you for your feedback. The manuscript has been thoroughly revised to improve the clarity, grammar, and overall quality of the English language.
After careful consideration, we have retained the original structure of the introduction as it ensures clarity, sets up the context and goals of the study as noted by Reviewer 2. However, we have revised the wordings to better align with Reviewer 3’s suggestions. |
|
Round 2
Reviewer 2 Report
Comments and Suggestions for Authors
Authors have ammended all suggestions and the manuscript is now fit for publication.
Author Response
Response to Reviewer 2 Comments
|
||
1. Summary |
|
|
Thank you very much for taking the time to review this manuscript and for your valuable suggestions, which have helped to improve its quality.
|
||
2. Questions for General Evaluation |
Reviewer’s Evaluation |
|
Does the introduction provide sufficient background and include all relevant references? |
Yes |
|
Are all the cited references relevant to the research? |
Yes |
|
Is the research design appropriate? |
Yes |
|
Are the methods adequately described? |
Yes |
|
Are the results clearly presented? |
Yes |
|
Are the conclusions supported by the results? |
Yes |
|
3. Point-by-point response to Comments and Suggestions for Authors |
||
Comments 1: Authors have amended all suggestions and the manuscript is now fit for publication. |
||
Response 1: Thank you for your positive and encouraging feedback. We are glad that the revised manuscript meets the expectations and is now considered suitable for publication.
4. Response to Comments on the Quality of English Language |
||
Point 1: The English is fine and does not require any improvement. |
||
Response 1: Thank you for your positive feedback regarding the quality of the English language. |
Reviewer 3 Report
Comments and Suggestions for Authors
1. These descriptions require appropriate citations to support the statements. L64-72
2. Although temperature and relative humidity can be monitored using sensors, the authors did not explain how such narrow ranges were maintained in the greenhouse. Fans and water walls alone are insufficient to regulate temperature and humidity so precisely; this level of control typically requires the use of air conditioners and heaters.
3. While the authors specified the cabbage cultivar used in the study, they did not include both the scientific name and the cultivar in the manuscript.
4. The authors still have not clarified which biotype of Bemisia tabaci was used in the study.
5. In the Introduction, the authors state that the tomato lines are resistant to TYLCV, but the virus used in the study was TYLCTHV. It should be noted that TYLCV and TYLCTHV are classified as distinct species, not strains.
6. Although the authors provided the isolate name and GenBank accession numbers of TYLCTHV used in the study, this information is not included in the manuscript.
7. According to standard academic practice, the manuscript should include details of the PCR kit used (brand and manufacturer) as well as the sequences of the primers employed in the assays. L176-179
Author Response
Response to Reviewer 3 Comments
|
|
|||
1. Summary |
|
|
|
|
Thank you very much for taking the time to review our manuscript. We sincerely appreciate your valuable guidance in enhancing the quality of our manuscript. Please find below the detailed point-by-point responses. The corresponding revisions have been made and are highlighted using track changes in the re-submitted files.
|
|
|||
2. Questions for General Evaluation |
Reviewer’s Evaluation |
|
||
Does the introduction provide sufficient background and include all relevant references? |
Yes |
|
||
Is the research design appropriate? |
Can be improved |
Noted and made changes wherever required. |
|
|
Are the methods adequately described? |
Yes |
|
|
|
Are the results clearly presented? |
Yes |
|
|
|
Are the conclusions supported by the results? |
Yes |
|
|
|
Are all figures and tables clear and well-presented? |
Can be improved |
Noted and made changes wherever required. |
|
|
3. Point-by-point response to Comments and Suggestions for Authors |
|
|
|
|
Comments 1: These descriptions require appropriate citations to support the statements. L64-72 |
|
|
||
Response 1: : Thank you for your valuable suggestion. We agree with this comment and have added appropriate citations to support the statements. [“Resistance to TYLCD was found in several wild relatives of tomato, from which six virus resistance genes (Ty-1 to Ty-6) have been identified. These genes have been mapped in wild tomato species including S. chilense (Ty-1, Ty-3, Ty-4 and Ty-6), S.habrochaites syn. L. hirsutum (Ty-2), and S. peruvianum (ty-5). Ty-1 and Ty-2 genes express complete or nearly complete dominance, while Ty-3 shows partial dominance [9]. The ty-5 gene is recessively inherited and resulted from a loss-of-function mutation [8]. Ty-1 and Ty-3 are the primary resistance genes widely used in tomato breeding programs. The highest resistance as based on disease incidence and severity was provided by Ty-1/Ty-3 gene combination [9].” L68-L76] |
|
|||
Comments 2: Although temperature and relative humidity can be monitored using sensors, the authors did not explain how such narrow ranges were maintained in the greenhouse. Fans and water walls alone are insufficient to regulate temperature and humidity so precisely; this level of control typically requires the use of air conditioners and heaters. |
|
|||
Response 2: Thank you for pointing this out. Upon revisiting the data, we realized that we had inadvertently reported mean ± standard error (SE) in the manuscript. We have corrected the manuscript to report the mean temperature and relative humidity as mean ± SD, instead of mean ± SE, as previously noted: 26.7 ± 3.4 °C temperature; 69.1 ± 3.9% relative humidity. [“The experiment was performed under greenhouse conditions (26.7 ± 3.4 °C temperature; 69.1 ± 3.9% relative humidity) at WorldVeg in Tainan, Taiwan, from 23 February to 22 May 2024.” L130-L132] Comments 3: While the authors specified the cabbage cultivar used in the study, they did not include both the scientific name and the cultivar in the manuscript. Response 3: Thank you for highlighting this. We agree with the comment and have made the necessary corrections in the manuscript. The cabbage (Brassica oleracea var. capitata) variety used in the study is a commercial cultivar from Known-You Seed, named "Green Tide" variety. [“Healthy B. tabaci individuals (both Q and B biotypes) were obtained from the WorldVeg virology department and reared on cabbage plants (Brassica oleracea var. capitata; "Green Tide" variety) in insect cages in an insect-proof plastic house.” L148-L151] |
|
|||
Comments 4: The authors still have not clarified which biotype of Bemisia tabaci was used in the study. |
|
|||
Response 4: Thank you for bringing this to our attention. We used both Q and B biotypes of Bemisia tabaci in the study. Since both biotypes are present in Taiwan and no strict measure was taken to maintain a clean population, a mixture of Q and B biotypes were present. [“Healthy B. tabaci individuals (both Q and B biotypes) were obtained from the WorldVeg virology department” L148-L149] |
|
|||
Comments 5: In the Introduction, the authors state that the tomato lines are resistant to TYLCV, but the virus used in the study was TYLCTHV. It should be noted that TYLCV and TYLCTHV are classified as distinct species, not strains. |
|
|||
Response 5: Thank you for your observation. We agree with your comment and have made the recommended correction accordingly. The tomato lines presented in this article carry resistance markers that confer protection against a broad spectrum of viruses, including Tomato yellow leaf curl Thailand virus (TYLCTHV), which is predominant in Taiwan. We have clarified this distinction in the revised manuscript. [“The tomato lines carry resistance that confers protection against a broad spectrum of virus, including TYLCTHV, which is predominant in Taiwan.” L115-L116]. We have also incorporated necessary corrections in the title and throughout the manuscript where required. Comments 6: Although the authors provided the isolate name and GenBank accession numbers of TYLCTHV used in the study, this information is not included in the manuscript. Response 6: Thank you for pointing this out. We agree with this comment. Therefore, the suggested correction was done. In this experiment, we used the LJ3-5 isolate of tomato yellow leaf curl Thailand virus (TYLCTHV), originally collected from Kaohsiung, Taiwan. The complete genomic sequences of this isolate, including both DNA-A and DNA-B components, are publicly available in GenBank under accession numbers EF577266 and EF577267. [“ In this experiment, we used the LJ3-5 isolate of TYLCTHV, originally collected from Kaohsiung, Taiwan. The complete genomic sequences of this isolate, including both DNA-A and DNA-B components, are publicly available in GenBank under accession numbers EF577266 and EF577267.” L145-L148] |
|
|||
Comments 7: According to standard academic practice, the manuscript should include details of the PCR kit used (brand and manufacturer) as well as the sequences of the primers employed in the assays. L176-179 |
|
|||
Response 7: We are grateful for your suggestion. The suggested correction was done. For the PCR, we used Super-Therm Gold DNA Polymerase (JMR Holding Inc.) Cat. No.:JMR-851; 5u/ul) and Premix dNTP, 2.5mM from Protech Technology Enterprise Co. Ltd., Taiwan (#PT-8014). The primer sequence used was F: GGACATGCAGGTGAGGAGTCC; R: TTATACGGATGGCCGCTTT. [“0.6 μL of dNTPs (Premix dNTP, 2.5mM; Protech Technology Enterprise Co. Ltd., Taiwan), 0.1 μL each of forward (GGACATGCAGGTGAGGAGTCC) and reverse primers (TTATACGGATGGCCGCTTT), 0.2 μL of Taq DNA polymerase (Super-Therm Gold DNA Polymerase; JMR Holdings Inc.,UK), and 2.0 μL of DNA template.” L189-L193]
|
|
|||
4. Response to Comments on the Quality of English Language |
|
|||
Point 1: The English is fine and does not require any improvement. |
|
|||
Response 1: Thank you. We are glad the language meets the required standard. |
|
